# Developmental loss of MeCP2 from VIP interneurons impairs cortical function and behavior

James M Mossner[1], Renata Batista-Brito[1,2], Rima Pant[1], Jessica A Cardin[1,3]*

[1]Department of Neuroscience, Yale University, New Haven, United States; [2]Dominick P. Purpura Department of Neuroscience, Albert Einstein College of Medicine, Bronx, United States; [3]Kavli Institute for Neuroscience, Yale University, New Haven, United States

**Abstract** Rett Syndrome is a devastating neurodevelopmental disorder resulting from mutations in the gene *MECP2*. Mutations of *Mecp2* that are restricted to GABAergic cell types largely replicate the behavioral phenotypes associated with mouse models of Rett Syndrome, suggesting a pathophysiological role for inhibitory interneurons. Recent work has suggested that vasoactive intestinal peptide-expressing (VIP) interneurons may play a critical role in the proper development and function of cortical circuits, making them a potential key point of vulnerability in neurodevelopmental disorders. However, little is known about the role of VIP interneurons in Rett Syndrome. Here we find that loss of MeCP2 specifically from VIP interneurons replicates key neural and behavioral phenotypes observed following global *Mecp2* loss of function.

*For correspondence:
jess.cardin@yale.edu

**Competing interests:** The authors declare that no competing interests exist.

Rett Syndrome is a severe neurodevelopmental disorder caused by mutations in the gene encoding methyl-CpG binding protein 2 (*MECP2*). Children who are affected by Rett develop normally during the initial year of postnatal life but regress rapidly thereafter, showing loss of motor skills and language and developing cognitive impairments, ataxia, respiratory problems, and stereotyped hand movements (*Chahrour and Zoghbi, 2007*). Extensive previous work has shown that many Rett-associated phenotypes are replicated in *Mecp2* loss-of-function mouse models (*Chen et al., 2001*; *Guy et al., 2001*; *Shahbazian et al., 2002*; *Chao et al., 2010*; *Ito-Ishida et al., 2015*). Rett Syndrome is strongly associated with seizure (*Hagberg et al., 1983*; *Amir et al., 1999*; *Chahrour and Zoghbi, 2007*), suggesting a possible role for GABAergic dysregulation in the pathophysiology underlying these symptoms. Indeed, previous work in mice found that conditional mutations of *Mecp2* that are restricted to GABAergic neurons recapitulate most of the observed phenotypes in the mouse model (*Chao et al., 2010*; *Ito-Ishida et al., 2015*), whereas rescue of *Mecp2* solely in GABAergic neurons ameliorates many phenotypes (*Ure et al., 2016*). These findings suggest a key role for the dysregulation of inhibitory interneurons in Rett Syndrome.

One major challenge in exploring GABAergic dysfunction in Rett Syndrome is the diversity of inhibitory interneurons, which can be subdivided into distinct classes that have different physiology, synaptic targets, and molecular markers. GABAergic interneurons that co-express vasoactive intestinal peptide (VIP), a sparse population that inhibit other interneurons and pyramidal cells (*Pfeffer et al., 2013*; *Pi et al., 2013*; *Prönneke et al., 2015*; *Garcia-Junco-Clemente et al., 2017*; *Chiu et al., 2018*), are thought to regulate powerfully the state-dependent function of neural circuits in the cerebral cortex (*Lee et al., 2013*; *Fu et al., 2014*; *Kamigaki and Dan, 2017*). In recent work, we found that early perturbation of VIP interneuron function caused profound dysregulation of cortical development, leading to altered neural activity, sensory processing, plasticity, and behavior (*Batista-Brito et al., 2017*). VIP cells may thus play a crucial role in cortical circuit development and

mature function. However, nothing is known about the contribution of VIP interneurons to neurodevelopmental dysregulation in Rett Syndrome.

Using a mouse model, we generated conditional mutations of *Mecp2* in VIP interneurons and compared these (i) with a conditional pan-interneuron mutation using the Dlx5/6 promoter to drive embryonic deletion in three major interneuron classes (VIP, parvalbumin-expressing interneurons [PV], and somatostatin-expressing interneurons [SST]) and (ii) with two conditional mutations in discrete interneuron populations (PV, SST). To identify the distinct contributions of each interneuron class, we assayed mortality, cortical activity, locomotor and anxiety phenotypes, and social behavior. Loss of MeCP2 selectively from VIP interneurons replicated key physiological and behavioral phenotypes observed in the pan-interneuron Dlx5/6 mutants, including altered firing rates, disruption of high-frequency cortical local field potential (LFP) patterns, and loss of state-dependent modulation of cortical activity. VIP interneuron-specific mutants further phenocopied impairments in marble burying and social behavior observed in the Dlx5/6 mutants. Overall, our findings suggest an unanticipated role for VIP interneuron dysfunction in the *Mecp2* loss-of-function model of Rett Syndrome.

## Results

### MeCP2 expression in PV, SST, and VIP interneurons

To confirm that MeCP2 is expressed in three major populations of GABAergic interneurons, we co-stained sections of cortex from adult mice with antibodies for interneuron markers and MeCP2 (*Figure 1*, *Figure 1—figure supplement 1*). As reported previously (*Ito-Ishida et al., 2015*), nearly all PV and SST interneurons expressed MeCP2. In addition, ~80% of VIP interneurons expressed MeCP2, suggesting a previously unappreciated potential role for this signaling pathway in VIP interneuron development and function.

To identify the unique contributions of VIP interneurons to the neural and behavioral phenotypes observed following *Mecp2* deletion (*Chao et al., 2010*), we generated four lines of conditional deletion mice lacking MeCP2 specifically in VIP ($Vip^{Cre+/-}Mecp2^{f/y}$; VIP mutants), PV ($Pvalb^{Cre+/-}Mecp2^{f/y}$; PV mutants), or SST ($Sst^{Cre+/-}Mecp2^{f/y}$; SST mutants) interneurons, or in all three populations of GABAergic interneurons ($Dlx5/6^{Cre+/-}Mecp2^{f/y}$; Dlx5/6 mutants) (*Anderson et al., 1997*; *Zerucha et al., 2000*; *Monory et al., 2006*; *Wang et al., 2010*) by crossing $Mecp2^{f/f}$ animals (*Samaco et al., 2008*) to interneuron-specific Cre lines. In each case, conditional mutants were compared to Cre-negative $Mecp2^{f/y}$ controls. We first examined the efficacy of conditional removal of MeCP2 from targeted interneuron populations in the cortex of $Cre^{+}Mecp2^{f/y}$ animals, finding near-complete removal of MeCP2 expression in each of the targeted interneuron classes (*Figure 1C*).

*Mecp2* deletion from different interneuron populations differentially impacted survival. We found that the mean age of death in $Mecp2^{-/y}$ (71.5 ± 8.7 days; n = 13; p<0.00001; H = 57.06), Dlx5/6 mutants (118.5 ± 15.2 days; n = 10; p<0.00001; H = 45.63), and SST mutants (172.5 ± 12.2 days; n = 16; p<0.0001; H = 36.30), but not PV (493.5 ± 6.5 days; n = 18; p>0.99; H = −3.50) or VIP mutants (447.5 ± 52.5 days; n = 8; p>0.99; H = 2.20), was significantly decreased compared to that of $Mecp2^{f/y}$ controls (464.7 ± 19.9 days; n = 38). There was no difference in the mean age of death between control and wild-type animals (p>0.99; H = 1.82; Kruskal-Wallis test with Dunn's post-test; *Figure 1D*).

### Seizure incidence following Mecp2 mutation

Previous work identified a characteristic seizure phenotype resulting from *Mecp2* deletion in the brain (*Chao et al., 2010*) and found that loss of MeCP2 from SST-expressing cells may confer a late-onset tendency towards seizure (*Ito-Ishida et al., 2015*). We therefore evaluated the incidence of seizure in each of the three interneuron-specific *Mecp2* deletion lines from weaning through to late adulthood. We compared the impact of MeCP2 depletion from VIP, PV, or SST interneuron populations with simultaneous depletion from all three interneuron classes using the Dlx5/6^{Cre} line. We further compared the interneuron-specific mutation mice with $Mecp2^{f/y}$ littermate controls. To identify the relative impact of *Mecp2* loss of function in GABAergic cells compared to loss of function of *Mecp2* in all cells, we also compared seizure incidence in the interneuron-specific deletion mice with that in mice carrying a complete knockout of the *Mecp2* gene ($Mecp2^{-/y}$).

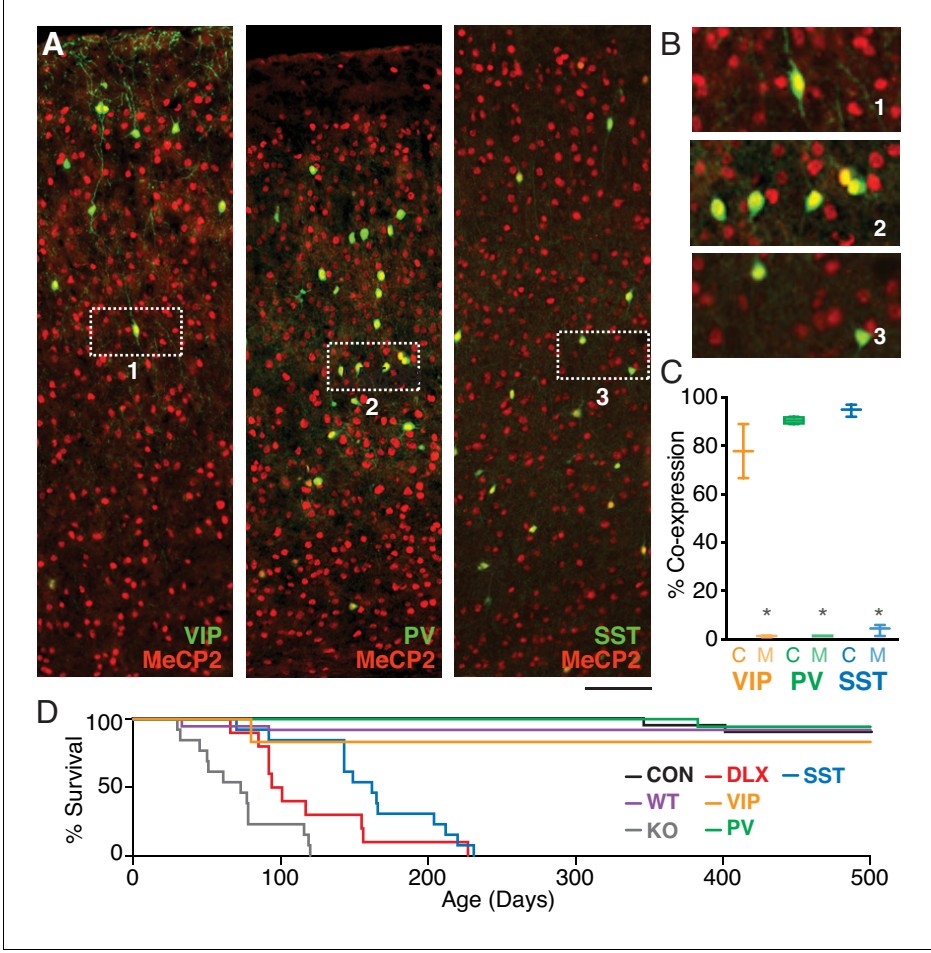

**Figure 1.** MeCP2 is expressed in three major GABAergic interneuron classes. (**A**) Co-staining for interneuron markers (green) and MeCP2 (red) reveals a high degree of co-expression in VIP (left), PV (middle), and SST (right) interneurons in the cortex. Scale bar denotes 100 µm. (**B**) Expanded view of insets 1–3 from panel (A). (**C**) Crossing interneuron-specific Cre lines with the conditional *Mecp2* line results in near-complete removal of MeCP2 expression from each target population in mutants (M) as compared to controls (C) (n = 4 mice per group). (**D**) Cumulative distribution plots of survival for controls (CON; black; n = 38), wild-types (WT; magenta; n = 10), and *Mecp2*$^{-/y}$ (KO; gray; n = 10), Dlx5/6 (DLX; red; n = 13), VIP (orange; n = 8), PV (green; n = 22), and SST (cyan; n = 36) mutants. *, p<0.05.

The online version of this article includes the following source data and figure supplement(s) for figure 1:

**Source data 1.** Statistical values for *Figure 1*, *Figure 1—figure supplement 1*, and *Figure 1—figure supplement 2*.

**Figure supplement 1.** MeCP2 expression in conditional deletion mice.

**Figure supplement 2.** Seizure incidence following conditional deletion of *Mecp2* from GABAergic interneurons.

MeCP2 depletion from specific interneuron populations had markedly different effects on seizure incidence (*Figure 1—figure supplement 2A-B*). We found that 100% of *Mecp2*$^{-/y}$ (n = 10) and Dlx5/ 6 mutant (n = 13) mice exhibited at least one seizure, compared to only 52.9% of SST mutants (n = 36), 35.0% of PV mutants (n = 22), and 37.5% of VIP mutants (n = 8). In comparison, the seizure rates in *Mecp2*$^{f/y}$ controls and in wild-types were 17.1% (n = 38) and 0% (n = 20), respectively, suggesting a small contribution of the floxed allele to the seizure phenotype (*Ito-Ishida et al., 2015*). The mean age of initial seizure was significantly earlier in *Mecp2*$^{-/y}$ mutants, but not the Dlx5/6, PV, SST, or VIP mutants, compared to *Mecp2*$^{f/y}$ controls (*Figure 1—figure supplement 2B*).

## Altered patterns of cortical and hippocampal activity following Mecp2 mutation

To examine the cellular and local network consequences of MeCP2 loss, we performed electrophysiological recordings in the cortex and hippocampus of awake animals. MeCP2 loss from interneurons caused alterations in the cortical LFP, a measure of local network activity (*Figure 2A–B*). MeCP2 depletion caused a modest change in LFP power, measured during periods of quiescence, in the 3–6 Hz range in the Dlx5/6 mutants (p=0.03; Kruskal-Wallis test with Dunn's post-test), but not in other groups (*Figure 2C*). However, we observed a robust broadband decrease in high-frequency LFP activity in the Dlx5/6 mutants that was replicated in the VIP mutants, but not PV or SST mutants (*Figure 2A*). Quantification of high-frequency activity around the gamma (40–55 Hz) band revealed a significant decrease in gamma-range activity in both the Dlx5/6 (p=0.005) and VIP mutants (p=0.04; Kruskal-Wallis test with Dunn's post-test; *Figure 2D*). We further found a loss of spike-field coherence in the gamma band in Dlx5/6 and VIP mutants (*Figure 2—figure supplement 1*). By contrast, hippocampal recordings revealed a loss of gamma-range LFP power in the Dlx5/6 mutants that was replicated by the SST mutants (*Figure 2—figure supplement 2*), suggesting potentially distinct cell-type-specific roles in different brain areas.

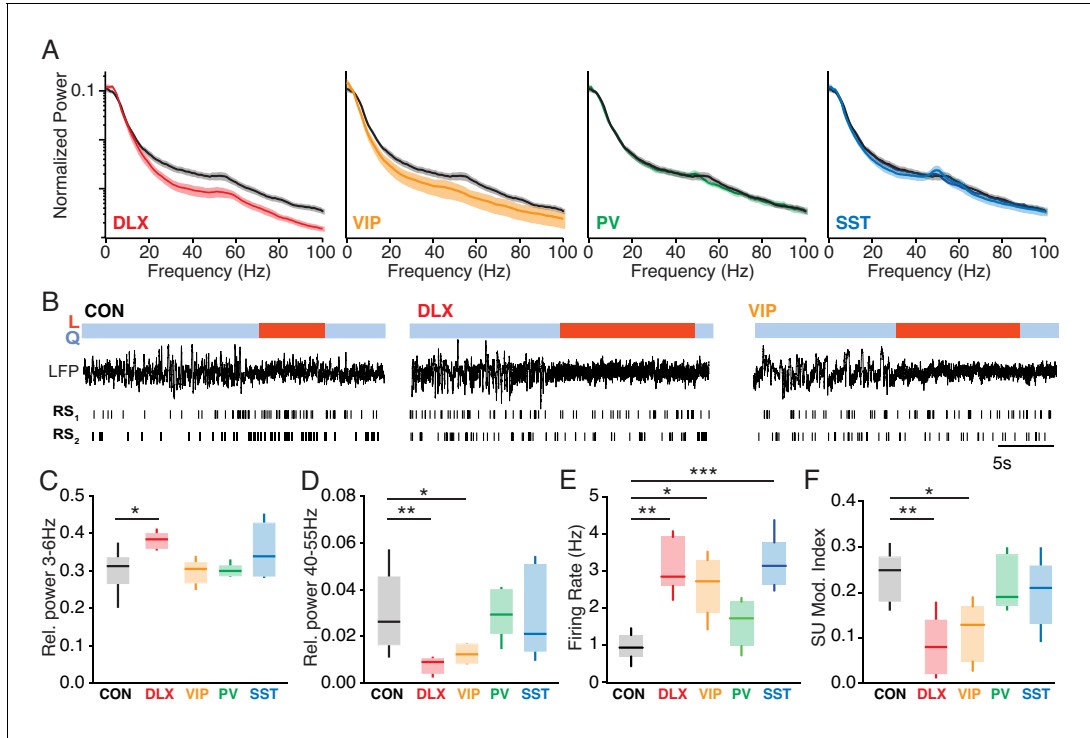

**Figure 2.** Altered state-dependent cortical activity following *Mecp2* deletion from VIP interneurons. (A) Population-averaged normalized cortical power spectra during quiescent sitting periods for Dlx5/6 mutants (red), VIP mutants (orange), PV mutants (green), and SST mutants (blue) compared to *Mecp2*$^{f/y}$ controls (black). (B) Example cortical LFP traces and raster plots for regular spiking (RS), putative pyramidal neurons during quiescence (Q, blue) and locomotion (L, red) periods in a *Mecp2*$^{f/y}$ control (CON), a Dlx5/6 mutant, and a VIP mutant. (C) Cortical relative LFP power in the 3–6 Hz band in during quiescent periods in *Mecp2*$^{f/y}$ controls (black; n = 12) and in Dlx5/6 (red; n = 5), VIP (orange; n = 6), PV (green; n = 6), and SST (cyan; n = 8) mutants. (D) Cortical relative LFP power in the 40–55 Hz band during quiescence. (E) Population-averaged single-unit firing rate of cortical RS cells during quiescence in *Mecp2*$^{f/y}$ controls (n = 7) and Dlx5/6 (n = 5), VIP (n = 5), PV (n = 5), and SST (n = 7) mutants. (F) Modulation of single-unit firing rate at locomotion onset in each group, measured as an index value. *, p<0.05; **, p<0.01; ***, p<0.001.

The online version of this article includes the following source data and figure supplement(s) for figure 2:

**Source data 1.** Statistical values for *Figure 2*, *Figure 2—figure supplement 1*, *Figure 2—figure supplement 2*, and *Figure 2—figure supplement 3*.
**Figure supplement 1.** Alteration in the temporal pattern of spiking.
**Figure supplement 2.** Altered hippocampal activity patterns in *Mecp2* mutants.
**Figure supplement 3.** No difference in neural activity among control groups.

Perturbation of interneurons during development can result in elevated firing rates due to loss of synaptic inhibition and the reorganization of neural circuits (*Close et al., 2012*; *Rossignol et al., 2013*; *Batista-Brito et al., 2017*). We therefore recorded cortical firing activity in awake mice with *Mecp2* mutations (*Figure 2B*). Single-unit recordings revealed that loss of MeCP2 in the Dlx5/6 mutants led to a three-fold increase in the firing rates of regular-spiking, putative excitatory pyramidal neurons as compared to that in control animals (p=0.004), and this finding was replicated in the VIP (p=0.03) and SST (p=0.002; Kruskal-Wallis test with Dunn's post-test) mutants (*Figure 2E*).

In previous work, we found that cortical firing rates are robustly modulated by changes in behavioral state, and are typically increased at the onset of locomotion (L) as compared to quiescence (Q) (*Vinck et al., 2015*). Loss of normal VIP interneuron activity reduces this state-dependent cortical modulation (*Fu et al., 2014*; *Batista-Brito et al., 2017*). To determine whether MeCP2 loss from VIP interneurons impairs this function, we examined state-dependent modulation in the *Mecp2* mutants. Both pan-interneuron MeCP2 loss in the Dlx5/6 mutants (p=0.004) and MeCP2 loss specific to VIP interneurons (p=0.02; Kruskal-Wallis test with Dunn's post-test), but not other interneuron

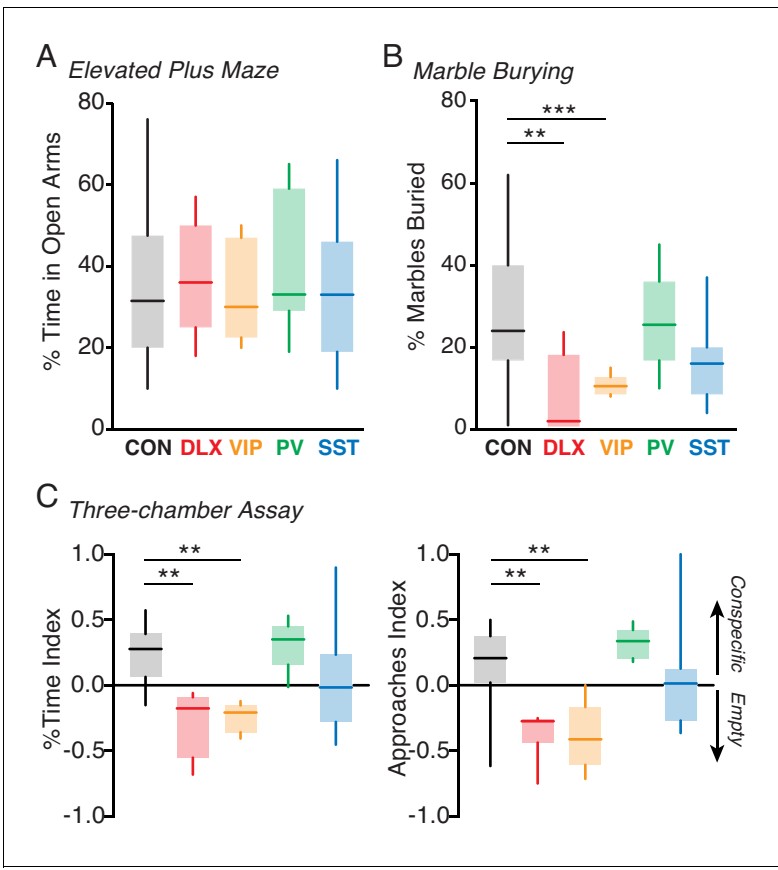

**Figure 3.** Loss of MeCP2 in VIP interneurons disrupts behavior. (**A**) Mean time spent in the open arms of the elevated plus maze for *Mecp2$^{f/y}$* controls (CON; black; n = 18) and Dlx5/6 (red; n = 5), VIP (orange; n = 5), PV (green; n = 7), and SST (cyan; n = 14) mutants. (**B**) Mean percentage of marbles buried by controls (n = 16) and by Dlx5/6 (n = 5), VIP (n = 8), PV (n = 6), and SST (n = 12) mutants. (**C**) Left: preference index for time spent with an unfamiliar conspecific in a small holding cage versus an empty cage for *Mecp2$^{f/y}$* controls (n = 16) and Dlx5/6 (n = 7), VIP (n = 6), PV (n = 5), and SST (n = 14) mutants. Right: preference index for approaches made to within 5 cm of the conspecific or the empty holding cage for each group. **, p<0.01; ***, p<0.001.

The online version of this article includes the following source data and figure supplement(s) for figure 3:

**Source data 1.** Statistical values for *Figure 3*, *Figure 3—figure supplement 1*, and *Figure 3—figure supplement 2*.

**Figure supplement 1.** Social behavior preferences are disrupted by *Mecp2* deletion from VIP interneurons.

**Figure supplement 2.** No difference in behavior among control groups.

populations, led to decreased state-dependent modulation of single-unit cortical firing rates, measured as a modulation index $[(FR_L–FR_Q)/(FR_L+FR_Q)]$ (*Figure 2F*). We did not find any differences in neural activity between $Mecp2^{f/y}$ controls, wild-types, and $Vip^{Cre}$ controls, suggesting that these results arise from cell type-specific *Mecp2* loss of function (*Figure 2—figure supplement 3*).

## Behavioral phenotypes associated with interneuron-specific *Mecp2* mutation

Previous work has linked *Mecp2* mutations in GABAergic interneurons to key impairments in motor, repetitive, and social behaviors (*Moretti et al., 2005*; *Chahrour and Zoghbi, 2007*; *Samaco et al., 2008*; *Chao et al., 2010*; *Kaufmann et al., 2012*; *He et al., 2014*; *Ito-Ishida et al., 2015*). We therefore examined the behavioral consequences of VIP-specific *Mecp2* mutations. In agreement with previous work (*Chao et al., 2010*; *Ito-Ishida et al., 2015*), we found no significant impairments in percentage of time spent in the open arms of the elevated plus maze task, a measure of anxiety (*Carobrez and Bertoglio, 2005*), for the Dlx5/6, VIP, PV, or SST mutants compared to controls (*Figure 3A*). Likewise, we found no impairment in locomotor behavior in the open field assay in any of the mutants (*Figure 3—figure supplement 1A*). By contrast, we found a significant impairment in marble burying, a measure of motor and repetitive behavior (*Thomas et al., 2009*; *Silverman et al., 2010*), in the pan-interneuron Dlx5/6 mutants compared to the controls (p=0.007). The deficit observed in the Dlx5/6 mutants was fully replicated in the VIP mutants (p=0.001; Kruskal-Wallis test with Dunn's post-test), but not in the PV or SST mutants (*Figure 3B*).

We tested social interaction behaviors in each *Mecp2* deletion line using the three-chamber sociability task (*Nadler et al., 2004*). Control animals exhibited a significant preference for a chamber containing a conspecific over an empty chamber (*Figure 3C*, *Figure 3—figure supplement 1B*). By contrast, the pan-interneuron Dlx5/6 mutants exhibited a reverse preference for the empty chamber (p=0.002). The VIP mutants (p=0.002; Kruskal-Wallis test with Dunn's post-test), but not the PV or SST mutants, fully replicated the effects of the pan-interneuron deletion, showing a significant preference for the empty chamber over the conspecific.

In addition to altered overall social preferences, *Mecp2* deletion affected the number of approaches mice made towards conspecifics. Control animals made more approaches to the conspecific than to the empty holding cage (*Figure 3C*). By contrast, the Dlx5/6 mutants made more approaches to the empty holding cage than to the conspecific animal (p=0.01). The VIP mutants (p=0.01; Kruskal-Wallis test with Dunn's post-test), but not the PV or SST mutants, fully replicated these effects of pan-interneuron deletion, approaching the empty holding cage more than the conspecific. Together, these data suggest that VIP interneurons may contribute to the deficits in social behavior caused by global *Mecp2* dysfunction. We did not find any differences in behavior between $Mecp2^{f/y}$ controls, wild-types, and $Vip^{Cre}$ controls (*Figure 3—figure supplement 2*), suggesting that these behavioral impairments arise as a consequence of cell-type-specific MeCP2 deletion.

## Discussion

Our results reveal an unanticipated role for VIP interneurons in the *Mecp2* loss-of-function model of Rett Syndrome. On the basis of previous characterizations of the *Mecp2* model (*Chahrour and Zoghbi, 2007*; *Samaco et al., 2008*; *Chao et al., 2010*; *He et al., 2014*; *Ito-Ishida et al., 2015*), we examined interneuron contributions to several major categories of neural dysregulation: mortality and seizure, cortical activity patterns, anxiety and repetitive behaviors, and social behavior. We found that *Mecp2* deletion from VIP interneurons recapitulates major phenotypes observed following pan-interneuron *Mecp2* deletion at the levels of both neural activity and behavior (*Supplementary file 1*).

Patients who have Rett Syndrome exhibit respiratory impairments and seizure (*Chahrour and Zoghbi, 2007*), and these phenotypes are replicated in mouse models following deletion of *Mecp2* from all GABAergic cells (*Chao et al., 2010*; *Ito-Ishida et al., 2015*). In agreement with previous work (*Ito-Ishida et al., 2015*), we found that *Mecp2* loss of function in SST interneurons conferred a late seizure phenotype. However, conditional mutations of *Mecp2* in VIP interneurons did not increase seizure or mortality rates. *Mecp2* knockout animals had significantly earlier seizure onset and mortality than the Dlx5/6 mutants, supporting previous findings that global loss of MeCP2 in excitatory neurons also contributes to seizure (*Goffin et al., 2014*; *Meng et al., 2016*) and that

respiratory impairments in the *Mecp2* knockouts may increase early mortality (*Chen et al., 2001*; *Guy et al., 2001*). In comparison, a previous study did not observe a seizure phenotype in Dlx5/6 mutants, possibly as a result of methodological differences (*Chao et al., 2010*). We observed a low rate of late-onset seizure in the *MeCP2^{f/y}* control animals, indicating that the decrease in MeCP2 levels associated with the conditional allele (*Samaco et al., 2008*) may be associated with epileptogenic consequences in addition to some mild behavioral phenotypes.

Recordings in the cortex of awake mice revealed a robust impact of *Mecp2* deletion on the pattern of cortical activity. We found a decrease in high-frequency LFP activity in the Dlx5/6 mutants that was replicated in the VIP mutants. In particular, VIP mutants exhibited a decrease in cortical gamma-range activity, which is associated with cognition and sensory processing (*Cardin, 2016*; *Cardin, 2018a*) and which may be impaired in Rett Syndrome (*Peters et al., 2015*). Although hippocampal gamma-range activity was also affected in the Dlx5/6 mutants, these effects were associated with SST rather than with VIP interneuron mutations, suggesting potential heterogeneity of the circuit-level impact of *Mecp2* deletion across brain areas. Loss of MeCP2 in the Dlx5/6 mutants was associated with a three-fold increase in cortical firing rates, and this increase was replicated in the VIP and SST mutants.

In the healthy cortex, transitions between behavioral states, such as quiescence and arousal or locomotion, are associated with robust modulation of cortical firing rates. (*Niell and Stryker, 2010*; *McGinley et al., 2015*; *Vinck et al., 2015*; *Tang and Higley, 2020*). However, loss of MeCP2 in the VIP and Dlx5/6 mutants caused a profound dysregulation of this state-dependent modulation. This loss of modulation is unlikely to result from a 'ceiling effect' caused by increased overall firing rates, as putative excitatory neurons in the SST and VIP mutants exhibited equally enhanced firing but only the VIP mutants showed a loss of state-dependent modulation. VIP interneurons are thought to play a key role in regulating the state-dependent modulation of cortical circuits, partly via strong inhibition of SST interneurons and consequent disinhibition of pyramidal neurons (*Pi et al., 2013*; *Fu et al., 2014*; *Prönneke et al., 2015*; *Karnani et al., 2016*; *Muñoz et al., 2017*). We had previously found that developmental perturbation of VIP interneurons by conditional deletion of the schizophrenia-associated gene *ErbB4* caused a similar loss of state-dependent cortical regulation (*Batista-Brito et al., 2017*). Together, these findings suggest that disruption of state-dependent cortical dynamics may be a common outcome of disease mechanisms affecting VIP interneurons.

Previous work has highlighted alterations in repetitive and motor behaviors, but not anxiety, following GABAergic deletion of *Mecp2* (*Chao et al., 2010*; *Ito-Ishida et al., 2015*; *Ure et al., 2016*). We therefore examined the impact of *Mecp2* deletion from VIP interneurons on locomotor and anxiety phenotypes. None of the *Mecp2* loss-of-function mutations resulted in anxiety-related or locomotor phenotypes in the elevated plus maze or the open field, respectively, in agreement with previous work (*Chao et al., 2010*; *Ito-Ishida et al., 2015*). However, the pan-interneuron Dlx5/6 mutants exhibited deficits in marble burying, a task that is susceptible to altered anxiety and OCD-like behaviors as well as changes in fine motor function (*Thomas et al., 2009*; *Silverman et al., 2010*). Cell-type-specific mutations of *Mecp2* in VIP interneurons, but not in the PV or SST populations, phenocopied this behavioral impairment.

Abnormal or reduced social behavior is a hallmark of many autism spectrum disorder models, and has previously been shown in mice lacking MeCP2 in all GABAergic populations (*Chao et al., 2010*; *Ito-Ishida et al., 2015*). We found that the pan-interneuron Dlx5/6 *Mecp2* mutants exhibited a reversal of normal social preferences in the three-chamber sociability assay, preferring an empty chamber to one containing a conspecific. Notably, *Mecp2* deletion from VIP, but not from PV or SST, interneurons fully replicated this phenotype. SST-specific deletion led to loss of any social preference, suggesting a potential contribution of both VIP- and SST-expressing cells to deficits in social behavior following global *Mecp2* loss of function. Together, these results suggest a previously unknown and potentially important role for VIP interneuron dysregulation in social behavior deficits in the *Mecp2*-deletion model.

Overall, our behavioral findings in the Dlx5/6, PV and SST mutants are in general agreement with previous work (*Supplementary file 1*). The *Pvalb^{Cre}* line used here largely expresses Cre recombinase in PV interneurons, along with some thalamocortical projection neurons (*Hippenmeyer et al., 2005*; *Cardin et al., 2009*). In comparison, the *Pvalb-2A-Cre* line used in some previous work (*Goffin et al., 2014*; *Ito-Ishida et al., 2015*) also expresses Cre in a subset of cortical pyramidal neurons and additional thalamic nuclei (*Madisen et al., 2010*). These differences may contribute to the

more severe behavioral phenotypes and early mortality previously observed in the *Pvalb-2A-Cre* line (*Ito-Ishida et al., 2015*). Other work examining the consequences of *Mecp2* deletion in the Pvalb$^{Cre}$ line likewise found only mild behavioral phenotypes (*He et al., 2014*). However, as the PV promoter only becomes active postnatally, our findings do not preclude a substantial contribution of embryonic *Mecp2* deletion from PV interneurons to Rett Syndrome phenotypes.

We find a unique impact of *Mecp2* deletion from VIP interneurons. Despite being few in number (*Rudy et al., 2011*), VIP interneurons are targets of multiple neuromodulatory systems and play critical roles in state-dependent regulation of local neural circuits (*Pi et al., 2013*; *Fu et al., 2014*; *Garcia-Junco-Clemente et al., 2017*; *Muñoz et al., 2017*), making them a potential key point of vulnerability in neurodevelopmental disease. Although our electrophysiology results are specific to cortex and hippocampus, the three interneuron classes examined here exhibit distinct cellular- and circuit-level properties and play key roles across many brain areas that contribute to behavior. In addition, dysregulation of one GABAergic population is probably amplified by extensive synaptic connectivity with other inhibitory interneuron classes (*Pfeffer et al., 2013*; *Cardin, 2018a*). Indeed, our previous work suggests that developmental disruption of VIP interneuron activity may have multiple circuit-level consequences, including loss of synaptic inhibition of other interneurons, altered experience-dependent plasticity, and dysregulated cortical circuit maturation, in addition to ongoing perturbation of normal adult function. *Mecp2* loss of function across multiple GABAergic interneuron classes may thus exert diverse influences on neural and behavioral deficits in Rett Syndrome.

## Materials and methods

### Animals

All experiments were approved by the Institutional Animal Care and Use Committee of Yale University. We used the *Dlx5/6$^{Cre}$* (JAX#008199; *Monory et al., 2006*), *Pvalb$^{Cre}$* (JAX#008069; *Hippenmeyer et al., 2005*), Sst$^{Cre}$ (JAX#013044; *Taniguchi et al., 2011*), and *Vip$^{Cre}$* (JAX#010908; *Taniguchi et al., 2011*) mouse lines to target all forebrain GABAergic interneurons, parvalbumin-expressing interneurons (PV), somatostatin-expressing interneurons (SST), and vasoactive intestinal peptide-expressing interneurons (VIP), respectively. We crossed each Cre line to the conditional *Mecp2* line (*Mecp2$^{f/f}$*; JAX# 007177; *Guy et al., 2001*). In each case, we assayed male mice that were hemizygous for the floxed *Mecp2* allele and heterozygous for Cre. All crosses were made on a C57BL/6J background (JAX#000664). Control animals were Cre-negative male mice that were hemizygous for the floxed *Mecp2* allele (*Mecp2$^{f/y}$*). We further compared *Mecp2$^{f/y}$* mice with age-matched wild-type C57Bl/6 mice (JAX#000664) and *Vip$^{Cre}$* mice (JAX#010908). In a subset of experiments, we compared the interneuron-specific crosses with male mice from the *Mecp2* knockout line (*Mecp2$^{-/y}$*; JAX#003890; *Guy et al., 2001*). All behavioral assays were performed at P120 except for those involving the *Dlx5/6$^{Cre+/-}$Mecp2$^{f/y}$* animals, which were assayed at P90 due to their early morbidity. All behavioral and electrophysiological assays were carried out in animals with no prior seizure incidence.

### Immunohistochemistry

For immunofluorescent staining of brain tissue, mice were perfused with 4% paraformaldehyde and post fixed for an hour before transferring into successive sucrose solutions at 15% and 30%. 20 μm thick cryosections were prepared for immunohistochemistry (IHC). Tissue was incubated with 1.5% normal goat serum (NGS) (Life Technologies) and 0.1% Triton X-100 (Sigma) in PBS for 60 min at room temperature. Sections were incubated with primary antibodies (Rat Anti-Somatostatin 1:250 [Millipore MAB354]; Anti-parvalbumin 1:1000 [Sigma P3088]; Anti-VIP 1:250 [ImmunoStar 20077]; Anti-MeCP2 1:250 [Millipore 07–013]) in the blocking buffer overnight at 4°C. After washing three times with buffer, sections were incubated with secondary antibodies for 1 hr at room temperature (secondary antibodies: Alexa Fluor 488, 594 or 647 [Life Technologies, 1:1000]). Finally, coverslips were mounted using ProLong Gold Mounting Medium with DAPI (Life Technologies) and imaged at 10x. Quantifications were performed in Adobe Photoshop. Pictures were divided into a grid measuring 1 × 1 mm in total and cells were counted in each grid square. The number of cells positive for antibody staining against MeCP2 was counted to assay the proportion of co-expressing cells.

### Headpost surgery and wheel training

For recordings performed in awake animals, mice were initially handled for 5–10 min/day for 5 days prior to a headpost surgery. On the day of the surgery, the mouse was anesthetized with isoflurane and the scalp was shaved and cleaned three times with betadine solution. An incision was made at the midline and the scalp resected to each side to leave an open area of skull. Two skull screws (McMaster-Carr) were placed at the anterior and posterior poles. Two nuts (McMaster-Carr) were glued in place over the bregma point with cyanoacrylate and secured with C&B-Metabond (Butler Schein). The Metabond was extended along the sides and back of the skull to cover each screw, leaving a bilateral window of skull uncovered over primary visual cortex. The exposed skull was covered with a layer of cyanoacrylate. The skin was then glued to the edge of the Metabond with cyanoacrylate. Analgesics were given immediately after the surgery and on the two following days to aid recovery. Mice were given a course of antibiotics (Sulfatrim, Butler Schein) to prevent infection and were allowed to recover for 3–5 days following implant surgery before beginning wheel training.

Once recovered from the surgery, mice were trained with a headpost on the wheel apparatus. The mouse wheel apparatus was 3D-printed (Shapeways Inc) in plastic with a 15 cm diameter and an integrated axle and was spring-mounted on a fixed base. A programmable magnetic angle sensor (Digikey) was attached for continuous monitoring of wheel motion. Headposts were custom-designed to mimic the natural head angle of the running mouse, and mice were mounted with the center of the body at the apex of the wheel. On each training day, a headpost was attached to the implanted nuts with two screws (McMaster-Carr). The headpost was then secured with thumb screws at two points on the wheel. Mice were headposted in place for increasing intervals on each successive day. If signs of anxiety or distress were noted, the mouse was removed from the headpost and the training interval was not lengthened on the next day. Mice were trained on the wheel for up to 7 days or until they exhibited robust bouts of running activity during each session. Mice that continued to exhibit signs of distress were not used for awake electrophysiology sessions.

### Locomotion detection

Wheel position was extracted from the output of a linear angle detector. We used a change-point detection algorithm that detected statistical differences in the distribution of locomotion velocities across time (see *Vinck et al., 2015*; *Batista-Brito et al., 2017*). Quiescent periods that lasted longer than 20 s were selected for analysis. For analysis of modulation with changes in behavioral state, we selected trials for which the preceding quiescent period lasted longer than 20 s, average speed until the next locomotion offset point exceeded 1 cm/s, and running lasted longer than 2 s.

### Extracellular recordings

LFP recordings were made with tetrodes (Thomas Recording GMBH, Germany) targeted to layers 2/3 and 5 of visual cortex and to the CA1 field of the dorsal hippocampus (AP: +1.5–2 mm; ML: 1.2–1.75, *Paxinos and Franklin, 2001*). Signals were digitized and recorded with a DigitalLynx 4SX system (Neuralynx, Bozeman MT). All data were sampled at 40 kHz and recordings were referenced to the cortical surface. LFP data were recorded with a bandpass 0.1–9000 Hz filter.

### Spike sorting

Spikes were clustered using previously published methods (*Vinck et al., 2015*; *Batista-Brito et al., 2017*). We first used the KlustaKwik 3.0 software (*Kadir et al., 2013*) to identify a maximum of 30 clusters using the waveform energy and the energy of the waveform's first derivative as clustering features. We then used a modified version of the M-Clust environment to separate units manually. Units were accepted if a clear separation of the cell relative to all the other noise clusters was observed, which generally was the case when isolation distance (ID) (*Schmitzer-Torbert et al., 2005*) exceeded 20 (*Vinck et al., 2015*). We further ensured that maximum contamination of the ISI (inter-spike-interval) histogram did not exceed 0.1% at 1.5 ms.

### Electrophysiology analysis

The firing rate was computed by dividing the total number of spikes a cell fired in a given period by the total duration of that period. To examine whether firing rates were significantly changed around locomotion onset, we computed the firing rate in the $[-0.5, 0.5]$ s window around locomotion onset

(L; as in *Vinck et al., 2015*) and compared this to the firing rate in the $[-5,-2]$ s quiescence (Q) period before locomotion onset by computing a modulation index ($[FR_L–FR_Q]/[FR_L+FR_Q]$). All LFP power analyses were made using data from quiescent periods after animals had been stationary for a minimum of 20 s and excluding data from within 10 s of the next locomotion bout. Relative power in the specified frequency bands was measured as a ratio between power in those bands and the total power. Spike-field coherence measures were performed as previously described (*Miri et al., 2018*), analysis code available at https://github.com/jesscardin/Miri-Vinck-et-al (*Cardin, 2018b*; copy archived at https://github.com/elifesciences-publications/Miri-Vinck-et-al). Power spectra were normalized to total power for visualization purposes.

## Seizure detection

All mice were handled for at least 10 min each day throughout the study. During the daily handling regime, the mice were assessed for seizures. If a seizure did occur, the mouse was immediately returned to its home cage. Seizure severity was scored using the Racine scale (1: Mouth and facial movement, 2: Head nodding, 3: Forelimb clonus, 4: Forelimb clonus and rearing, 5: Forelimb clonus, rearing, and falling). Seizures were defined as events reaching Racine scale levels 4 or 5, with animals exhibiting rearing and forelimb clonus or rearing, forelimb clonus, and falling.

## Morbidity analysis

After weaning at P21, all mice were monitored every day throughout the study. All deaths were noted, and animals were tracked daily until P500.

## Behavioral analysis

The elevated plus maze, marble-burying, and sociability assays were performed under low-level (20–25 lux) illumination. In each assay, mice were given 15 min to acclimate to the behavioral assay room. In all cases, the researcher was blind to the genotypes of the mice until after all behavioral data were scored.

### Elevated plus maze

Custom Labview software was used to control a camera recording the mouse's locomotion in the maze. At the beginning of the session, the mouse was placed at the center of the maze and allowed to freely move on either arm for five minutes. At the end of the session, the mouse was returned to the home cage and the maze was cleaned for the next mouse. Video recordings of mouse behavior were hand-scored to determine the amount of time spent in the open and closed arms of the maze.

### Open field

The open field assay was performed in a 30 cm square box divided into nine quadrants. Custom software (Labview) was used to control a camera recording the mouse's path in the box. At the start of the session, the mouse was placed in the center quadrant of the box and allowed move freely for 20 min. After the time elapsed, the mouse was returned to the home cage and the box was cleaned for the next mouse. ImageJ software was used to analyze the total distance traversed by the mouse.

### Marble burying

12 marbles were evenly placed in a cage with 1 inch of clean bedding. Custom Labview software was used to control a camera recording the mouse's activity in the cage. The mouse was placed in the center of the cage with the marbles and allowed to explore the cage for 20 min. At the end of the session, the mouse was returned to the home cage and the marbles were cleaned with a 10% bleach solution.

The proportion of marbles buried was analyzed in ImageJ using the 'analyze particles' function to compare the initial and final exposed surface area of the marbles.

### Sociability

The sociability apparatus was divided into three equal areas with Plexiglas dividers, each with an opening allowing access to neighboring chambers. Custom Labview software was used to control a camera that recorded the mouse's activity in the chamber. An unfamiliar age- and sex-matched

conspecific mouse (reared in separate cages, C57BL/6 genotype) was placed into a small cylindrical holding cage in one side of the chamber and an identical empty holding cage was placed in the other side. The location of the conspecific was randomly varied across trials. At the beginning of the session, the test mouse was placed in the central chamber and habituated to the central chamber for ten minutes. The dividers were then removed to allow the mouse to move freely among all the chambers for ten additional minutes. At the end of the session, the mice were returned to their home cages and the apparatus was cleaned. Video recordings of the mouse's behavior were scored to determine the amount of time spent in each of the three partitions and the number of approaches that the test mouse made to the conspecific and the empty holding cage. An approach was defined as the test mouse coming within a 5-cm radius of a cage or making contact with a holding cage. Social preferences were calculated both as comparisons of raw values and as index values for time spent ($[Time_C-Time_E]/[Time_C+Time_E]$) and approaches ($[App_C-App_E]/[App_C+App_E]$), where C denotes conspecific and E denotes the empty container.

### Statistical analysis

Paired and unpaired non-parametric tests generated in GraphPad Prism (version 8 for Mac; San Diego CA) were used throughout the study to accommodate non-normal data distributions. Animals were used as the 'n' in all analyses. Exact p values and estimation statistics are reported in the source data files for all tests. All group data are shown as box-and-whisker plots in which the bars denote the minimum and maximum of the distribution and the box denotes the first and third quartiles and the median.

## Acknowledgements

This work was funded by a Simons Foundation SFARI Pilot grant to JAC and NIMH R01 MH113852. RBB was supported by a NARSAD Young Investigator Award. The authors thank Dr Michael J Higley for feedback on the manuscript, Ms Victoria Hernandez and Mr Brandon Wanke for assistance in performing behavioral assays, and the lab of Dr Jane Taylor for help with behavioral equipment.

## Additional information

### Funding

| Funder | Grant reference number | Author |
|---|---|---|
| Simons Foundation | SFARI | Jessica A Cardin |
| National Institute of Mental Health | MH113852 | Jessica A Cardin |
| Brain and Behavior Research Foundation | NARSAD Young Investigator Award | Renata Batista-Brito |

The funders had no role in study design, data collection and interpretation, or the decision to submit the work for publication.

### Author contributions

James M Mossner, Formal analysis, Writing - original draft; Renata Batista-Brito, Data curation, Formal analysis, Visualization, Methodology, Writing - original draft; Rima Pant, Formal analysis, Visualization, Methodology; Jessica A Cardin, Conceptualization, Data curation, Formal analysis, Supervision, Funding acquisition, Visualization, Writing - original draft, Writing - review and editing

### Author ORCIDs

Jessica A Cardin  https://orcid.org/0000-0002-8209-5466

### Ethics

Animal experimentation: This study was performed in strict accordance with the recommendations in the Guide for the Care and Use of Laboratory Animals of the National Institutes of Health. All

experiments were approved by the Institutional Animal Care and Use Committee of Yale University (#11317).

### Decision letter and Author response

Decision letter https://doi.org/10.7554/eLife.55639.sa1
Author response https://doi.org/10.7554/eLife.55639.sa2

## Additional files

### Supplementary files

• Supplementary file 1. Consequences of cell-type-specific *Mecp2* deletion. Summary table of phenotypes observed in conditional deletion mice with loss of *Mecp2* function in GABAergic interneurons (upper) and glutamatergic excitatory neurons (lower). Observed phenotypes are noted as Y/N, assays that were not performed in a given study are left blank. The two distinct *Pvalb*$^{Cre}$ mouse lines used by different studies are identified by their JAX line numbers.

• Transparent reporting form

### Data availability

Source data files are included for each figure and supplementary figure. Analysis code is available at https://github.com/jesscardin/Miri-Vinck-et-al (copy archived at https://github.com/elifesciences-publications/Miri-Vinck-et-al). All data included in this study will be freely available upon request, as the data files and associated intermediate analysis files are very large (400GB) and depositing the full data is not feasible.

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
