## [Decision Letter]

**Acceptance summary:**

In this manuscript, Mossner et al. perform a detailed comparison of the effects of deletion of the gene MeCP2 from different subsets of GABAergic interneurons in mice. They find that MeCP2 loss from VIP interneurons recapitulates several phenotypes observed following pan-interneuron MeCP2 deletion at the level of both neural activity and behaviour. In many cases these phenotypic disruptions were distinct, and not observed when MeCP2 was lost in SST and PV interneurons. This points to MeCP2 loss in VIP interneurons likely playing an important role in the pathophysiology of Rett syndrome.

**Decision letter after peer review:**

Thank you for submitting your article "Developmental loss of MeCP2 from VIP interneurons impairs cortical function and behavior" for consideration by *eLife*. Your article has been reviewed by three peer reviewers, one of whom is a member of our Board of Reviewing Editors, and the evaluation has been overseen by John Huguenard as the Senior Editor.

The reviewers have discussed the reviews with one another and the Reviewing Editor has drafted this decision to help you prepare a revised submission.

Overall, all three reviewers were enthusiastic about the study, and commended the authors on the thorough side by side comparison of the consequences of MeCP2 loss in various neuronal subtypes, with a specific focus on MeCP2 loss in VIP interneurons.

Summary:

In this manuscript, "Developmental loss of MeCP2 from VIP interneurons impairs cortical function and behavior," Mossner et al. compare the effects of deletion of the only copy of MeCP2 in males from different subsets of GABAergic interneurons. They take advantage of the existing floxed MeCP2 mouse line, crossing it with PV-Cre, SST-Cre, and VIP-Cre mouse lines to delete the gene in three major classes of interneurons, and compare the results with a simultaneous knockout from all three classes (*Dlx5/6^Cre^*) and whole brain knockout. They compare all these mice on measures of mortality, cortical function, and behavior. They focus particularly on the VIP results because, the effects of VIP-specific knockout of MeCP2 have not been previously reported. They find that MeCP2 loss from VIP interneurons recapitulates several phenotypes observed following pan-interneuron MeCP2 deletion at the level of both neural activity and behaviour. In many cases these phenotypic disruptions were distinct, and not observed when MeCP2 was lost in SST and PV interneurons. This points to MeCP2 loss in VIP neurons likely playing an important role in the pathophysiology of Rett syndrome.

Essential revisions:

1) The major experimental revision we request are additional controls with VIP-Cre mice focussing on assays in which you found differences with the VIP knockout mice (excluding life span/ mortality as we realise this would take significant time). That is, we request that you demonstrate that the VIP neuron knockout phenotypes observed are not due to the VIP-Cre line itself.

---

## [Author Response]

Essential revisions:1) The major experimental revision we request are additional controls with VIP-Cre mice focussing on assays in which you found differences with the VIP knockout mice (excluding life span/ mortality as we realise this would take significant time). That is, we request that you demonstrate that the VIP neuron knockout phenotypes observed are not due to the VIP-Cre line itself.

We have added new data to address the potential contribution of the VIP-Cre line to the observed phenotypes. To validate the results in Figure 2, we have added electrophysiology data comparing control (*MeCP2^f/y^*), wild-type (C57Bl/6), and *VIP^Cre+/-^* animals. We find no difference between these three groups for LFP, firing rates, or modulation index (Figure 2—figure supplement 3). To similarly validate the results in Figure 3, we have added behavioral data for the marble burying task and sociability assay. We find no difference between the controls, wild-types, and VIP-Cre mice for either assay (Figure 3—figure supplement 2). Together, these new data suggest that neither the electrophysiological nor the behavioral phenotypes we observe in the *MeCP2^f/y^VIP^Cre+/-^* mutants can be attributed to the VIP-Cre line.